# Timing of Novel Drug 1A-116 to Circadian Rhythms Improves Therapeutic Effects against Glioblastoma

**DOI:** 10.3390/pharmaceutics13071091

**Published:** 2021-07-16

**Authors:** Laura Lucía Trebucq, Georgina Alexandra Cardama, Pablo Lorenzano Menna, Diego Andrés Golombek, Juan José Chiesa, Luciano Marpegan

**Affiliations:** 1Laboratorio de Cronobiología, Universidad Nacional de Quilmes-CONICET, Bernal 1876, Buenos Aires, Argentina; laura.trebucq@gmail.com (L.L.T.); dgolombek@unq.edu.ar (D.A.G.); 2Laboratorio de Oncología Molecular, Universidad Nacional de Quilmes-CONICET, Bernal 1876, Buenos Aires, Argentina; gcardama@gmail.com (G.A.C.); plmenna@unq.edu.ar (P.L.M.); 3Departamento de Física Médica, Comisión Nacional de Energía Atómica, Bariloche 8400, Río Negro, Argentina

**Keywords:** chronopharmacology, Rho GTPase, glioma, brain tumor

## Abstract

The Ras homologous family of small guanosine triphosphate-binding enzymes (GTPases) is critical for cell migration and proliferation. The novel drug 1A-116 blocks the interaction site of the Ras-related C3 botulinum toxin substrate 1 (RAC1) GTPase with some of its guanine exchange factors (GEFs), such as T-cell lymphoma invasion and metastasis 1 (TIAM1), inhibiting cell motility and proliferation. Knowledge of circadian regulation of targets can improve chemotherapy in glioblastoma. Thus, circadian regulation in the efficacy of 1A-116 was studied in LN229 human glioblastoma cells and tumor-bearing nude mice. Methods. Wild-type LN229 and BMAL1-deficient (i.e., lacking a functional circadian clock) LN229E1 cells were assessed for rhythms in TIAM1, BMAL1, and period circadian protein homolog 1 (PER1), as well as Tiam1, Bmal1, and Rac1 mRNA levels. The effects of 1A-116 on proliferation, apoptosis, and migration were then assessed upon applying the drug at different circadian times. Finally, 1A-116 was administered to tumor-bearing mice at two different circadian times. Results. In LN229 cells, circadian oscillations were found for BMAL1, PER1, and TIAM1 (mRNA and protein), and for the effects of 1A-116 on proliferation, apoptosis, and migration, which were abolished in LN229E1 cells. Increased survival time was observed in tumor-bearing mice when treated with 1A-116 at the end of the light period (zeitgeber time 12, ZT12) compared either to animals treated at the beginning (ZT3) or with vehicle. Conclusions. These results unveil the circadian modulation in the efficacy of 1A-116, likely through RAC1 pathway rhythmicity, suggesting that a chronopharmacological approach is a feasible strategy to improve glioblastoma treatment.

## 1. Introduction

The mammalian circadian system is composed of a hierarchical arrangement of oscillators, with a master pacemaker located in the hypothalamic suprachiasmatic nuclei (SCN) [1]. These nuclei control the rhythmicity of most physiological and behavioral processes, enabling their entrainment to daily environmental variables such as light-dark and temperature cycles [2]. The core molecular components of circadian oscillations are based on transcriptional-translational auto-regulatory feedback loops generated by the activity of circadian genes of which bmal1, period (1 and 2), cry (1 and 2) and clock are the main components. Because circadian oscillations are produced by an endogenous mechanism, they persist in the absence of environmental cycles and even in isolated cells and tissues in vitro. Peripheral oscillators capable of generating circadian cycles are present in most cells and tissues in the body and are coordinated by the SCN rhythmic neuroendocrine (e.g., melatonin, glucocorticoids), autonomic and behavioral outputs. These physiological oscillations lead to daily rhythms in pharmacokinetics (e.g., absorption, metabolism) and pharmacodynamics (interaction with effector targets) [3,4] of many drugs, providing the basis for chronopharmacology, which seeks for the circadian time with maximum efficacy of treatments and minimal side effects [5].

Glioblastomas (GBM) are the most common and malignant primary tumors of the central nervous system in adults and are composed of poorly differentiated astrocytic cells and their main characteristics are exhaustive proliferation, infiltration and resistance to apoptosis [6,7]. The current treatment for high-grade gliomas includes surgery, radiotherapy and chemotherapy with temozolomide (TMZ) [8] but even when treated, the average survival time is just over one year [9]. Considering that the inclusion of TMZ in the treatment of GBM patients increased their life expectancy by just a few months, neither the quality nor the survival of the patients have improved considerably in the last three decades [10]. Consequently, new therapeutic approaches are essential for GBM patients worldwide [11,12]. The 1A-116 compound was recently proposed as a novel drug for the treatment of glioblastoma and other tumors [13,14]. Both in LN229 and U-87 glioma cell lines, it specifically inhibited the activation of RAC1 [13], a monomeric GTPase involved in a variety of cellular processes required for tumor progression (proliferation, migration, and cytoskeleton reorganization) [15,16]. 1A-116 was designed to block the binding of RAC1 to guanine exchange factors (GEFs) such as Tiam1, by interacting with the Trp56 residue on its surface. As with many other drugs (including nitrates [17], benzodiazepines [18], antidepressants [19] and those to treat essential hypertension [20]), the effectiveness of 1A-116 could be further improved by finding the best time for delivery. The improved efficacy of chronopharmacological approaches for over thirty anti-cancer drugs was assessed recently in pre-clinical studies and clinical trials, which showed that controlling administration time leads to lower toxicity [21,22,23,24,25]. However, to the best of our knowledge, only one chronopharmacological work was published reporting circadian modulation of the effects of TMZ in patients [26]. To improve life expectancy and quality for GBM patients, it is critical to identify new druggable targets and to optimize treatment schedules. In this context, the aim of this work was to assess the circadian modulation of the efficacy of 1A-116 acting on human GBM LN229 cell lines in vitro and in vivo in LN229 xenografts implanted in nude mice.

## 2. Materials and Methods

### 2.1. Cell Lines

The established human glioma cell lines, LN229 (American Type Culture Collection (ATCC) CRL-2611), were derived from grade IV astrocytomas and were maintained according to ATCC recommendations. They were grown at 37 °C, under a 5% CO_2_ and 100% humidity atmosphere. The culture medium used in the growth contained 10% fetal bovine serum (FBS) in Dulbecco’s Modified Eagle’s Medium (DMEM, Gibco, Thermo Fischer Scientific, Waltham, MA, USA) and was supplemented with glutamine (2 mM). Confluent cell cultures were then subcultured twice a week, using standard procedures.

Circadian synchronization of cell cultures was achieved by following the procedures outlined in [27]; by exposing cells to a two-hour serum shock (50% FBS in DMEM), followed by regular DMEM. After the serum shock, the elapsed time (in hours) was recorded as the hours post-synchronization (HPS) and was used to set the circadian time. Only the cells with passage numbers between 250 and 280 were used.

The genotype was confirmed with PCR and sequencing (Productos Bio-Lógicos, Buenos Aires, Argentina) and Short Tandem Repeats (STR) (Easy DNA, Buenos Aires, Argentina). During PCR and sequencing, the following oncogenes were looked at: PTEN wt, p53 mutation Pro98Leu and p14 ARF and p16 deletions (Appendix A). Deletions of p14 and p16 were confirmed by PCR and the sequences of PTEN and P53 were found as expected. The STR profile was performed according to ATCC recommendations and it reported the following results: Amelogenin: X, CSF1PO: 12, D13S317: 10,11, D16S539: 12, D5S818: 11,12, D7S820: 8,11, TH01: 9.3, TPOX: 8, vWA: 16,19; which are consistent with the LN229 STR profile displayed in Appendix A. Finally, the cell line was tested for mycoplasma contamination twice a year. 

### 2.2. Drugs and Reagents

1A-116 (MW 307.6 g/mol, 99.3% pure) was provided by Chemo-Romikin S.A (Buenos Aires, Argentina). Unless stated otherwise, all reagents were provided by Sigma-Aldrich (St. Louis, MO, USA).

### 2.3. CRISPR/cas9 Knock-Down of Bmal1 Expression

Knock-down experiments were performed as previously described [28]. Briefly, LN229 cells were cultured in 12-well plates and transfected with StBL3E1 Crispr-Cas9 for bmal1 human knock-down (kindly provided by Dr. Mario Guido). Transfection was performed using Lipofectamine 3000 (Thermo Fisher Scientific, Waltham, MA, USA) according to manufacturer’s recommendations, in serum and antibiotics-free DMEM medium. After 6 h, the medium was replaced with fresh complete DMEM, and after 72 h, the cells were treated with 1 μg/mL puromycin for 72 h to positively select the transfected cells. The cells were subcultured and 1 μg/mL puromycin was added every ten passages to ensure the selection of transfected cells. The pool of selected LN229 cells was labeled as LN229E1. The successful bmal1 knock-down was validated for LN229E1 using the following specific primers: Fw primer: 5′-CAACGTGCCATGTGTTA-3′, Rv primer: 5′-GAAGGCCCAGGATTCCA-3′, for amplification of a ~300 pb fragment of exon 2 of bmal1, flanking the CRISPR recognition site, followed by Sanger sequencing (Macrogen, Seoul, Korea). Sequences of amplicons were aligned with bmal1 sequence using software nBLAST (version 2.11.0, NCBI, Bethesda, MD, USA), confirming the presence of edited exon 2 in LN229E1 (Appendix A). To assess if the disruption of the bmal1 gene promotes a non-circadian phenotype, the loss of circadian rhythms of BMAL1 and PER1 clock proteins in synchronized cultures was measured by in-cell western assay [29] and real-time bioluminescent recording of bmal1 gene promoter activity with pGL3-Bmal1-dLuc plasmid.

### 2.4. Bioluminescent Recordings

A total of 2.5 × 10^5^ LN229 and LN229E1 cells/well were plated in 35 mm plates (Cell Star, Greiner Bio-One, Frickenhausen, Germany). Similarly, transfection with pGL3-Bmal1-dLuc plasmid [30] was performed using Lipofectamine 3000 according to manufacturer’s recommendations, in a serum- and antibiotics-free DMEM medium. After 6 h, the medium was replaced with fresh complete DMEM, followed by serum shock synchronization. The DMEM was replaced 2 h later with a medium that was suitable for long-term bioluminescence recording: 1% FBS in DMEM without phenol red, supplemented with HEPES (Gibco, Life Technologies, Waltham, MA, USA), glutamine (2 mM), pyruvate (2 mM) and luciferin (0.1 mM). Finally, the plates were sealed with silicone grease to avoid medium evaporation.

Bioluminescence was measured with a luminometer (Kronos Dio, ATTO, Tokyo, Japan) at 37 °C for a total time of 72 h. Readings were recorded every 30 min, each with an integration time of 6 min. The data were then analyzed for circadian periodicity using the ChronoStar 2.0 software (Stephan Lorenzen, Hamburg, Germany) [31].

### 2.5. Cell Proliferation Assay

A total of 3 × 10^3^ cells/well were plated in 96-well plates (Cell Star, Greiner Bio-One, Germany). To characterize the circadian modulation of the response to 1A-116, synchronized cultures were treated during 72 h with 20 μM 1A-116, or vehicle, delivered at 0, 5, 10, 15, 20, or 25 HPS (5 h intervals during 25 h). At 72 h, the cultures should form confluent monolayers at standardized culture growth conditions. Cell growth (G) was assessed by measuring the uptake of either crystal violet 0.1% (measuring absorbance at 570 nm), or Calcein AM (Life Technologies, Carlsbad, CA, USA), measuring fluorescence emission at 520 nm. Uptake in drug-treated cultures was normalized dividing it by the average uptake in vehicle-treated cultures. Inhibition of proliferation (Ip) was quantified as Ip = 1 − G. Three independent assays were performed for each experimental condition.

Using the same assay, dose-response experiments were performed to calculate the half-maximal inhibitory concentration (IC50). Experiments were performed including LN229 and LN229E1 with 1A-116 at 10, 20, 40, 50, or 100 μM at 10 and 23 HPS, which are respectively the circadian time of maximum and minimum Ip, found in the previous experiments. The collected data were processed and fitted to dose-response curves generated by non-linear fitting using the variable Hill slope Dose-Response function within OriginPro9 (OriginLab Corporation, Northampton, MA, USA).

### 2.6. Cell Viability Assay

To determine the acute effects on cell viability, 2.5 × 10^3^ cells/well were plated in 96-well plates (Cell Star, Greiner Bio-One, Frickenhausen, Germany). Following synchronization, cells were treated with either 1A-116 at 5 or 10 μM, or vehicle at 10 or 23 HPS. Because Calcein AM stains only living cells, cell viability was measured as the number of fluorescent cells determined 16 h after the drug treatment. Three independent assays were performed for each experimental condition.

### 2.7. Cell Migration Assays

2.5 × 10^5^ LN-229 cells were plated and synchronized by serum shock in 12-well plates having a silicon 3.5 mm elastomer (Sylgard 184) stopper previously adhered to the center of each well. Except for technical controls (t0), the silicon stoppers in all wells were removed at the time of treatment to generate a cell-free area into which the surrounding cells could migrate. Cells were treated with 1A-116 (10 μM) or vehicle at 10 HPS or 23 HPS, and cell migration was measured 16 h later (as stated in [13]) after labeling cells with Calcein AM. Because 1A-116 has effects on proliferation and apoptosis, the time allowed for the cells to migrate was enough for them to invade the cell-free area while keeping the number of cell divisions at a minimum. The stoppers in the three t0 wells were removed 1 h before quantification, the results were averaged and subtracted from the measurements of all other wells to account for the cells that could stochastically detach and fall in the cell-free area. Images of the cultures were obtained using a Cytation 5 Imaging Reader (Biotek Instruments, Winooski, VT, USA) with a 2.5x objective and GFP (488/520 nm) channels. Two regions of interest within the same area were selected and measured in each image, one in the center of the cell-free area and the other one outside, where the cell monolayer was intact (no stopper). This allowed measuring the cells in the migration area relative to the cells outside, accounting for possible effects of the number of cells available to enter the migration area. The number of labeled cells inside the migration area and the total covered migration area for each treatment were assessed using ImageJ software (version 1.53a, Wayne Rasband, Stapleton, NY, USA). The results shown correspond to the average of three independent experiments.

### 2.8. Early Apoptosis Assay

In total, 3 × 10^3^ LN-229 cells/well were plated in 96-well plates. After 24 h, the cells were synchronized and treated for 6 h (as stated in [13]) with either 20 and 50 μM 1A-116, or vehicle at 10 HPS and 23 HPS. Early apoptosis was measured using fluorescent-labeled Annexin V; a phospholipid-binding protein with a high affinity for phosphatidylserine that is present in the outer surface of the cell membrane in apoptotic cells. The Alexa Fluor 488 Annexin V kit (Molecular Probes, Thermo Fisher Scientific, Waltham, MA, USA) was used according to the manufacturer’s instructions. Each measurement was performed using a Cytation 5 Imaging Reader (Biotek Instruments, Winooski, VT, USA) at 10x objective and DAPI (360/460 nm) and GFP channels.

### 2.9. Quantitative Real-Time RT-PCRs

The 8 × 10^5^ LN-229 cells were plated in 6-well plates and were synchronized at 0, 4, 8, 12, 16, 20 and 24 HPS (4 h intervals during 24 h). The total RNA was extracted using the EasyPure RNA kit (Transgen Biotech, Beijing, China) according to the manufacturer’s instructions. Next, the DNA was removed from all samples using DNAsa I free Rnasa (Promega, Madison, WI, USA). Reverse transcriptase Superscript II (Thermo Fisher Scientific, USA) was used according to the manufacturer’s instructions to then synthesize cDNA. Quantitative real-time RT-PCR (qPCR) analysis was performed using the Power Sybr Green PCR (Thermo Fisher Scientific, USA) with a Step-One Real-time PCR system (Thermo Fisher Scientific, Waltham, MA, USA), using the following primers: Rac1, Fwd: 5′-GTGCAGACACTTGCTCTCCT-3′, Rv: 5′-AATGGCAACGCTTCATTCGG-3′, Tiam1: Fwd:5′-TGCCGTGTTCTGACTTACC-3′, Rv: 5′-ACATGAATCGCCACCCTCTC-3′, Bmal1:Fwd: 5′-CCACTGTTCCAGGGATTCCA-3′, Rv: 5′-GGAGGCGTACTCGTGATGTT-3′, Actin: Fwd: 5′-GGACTTCGAGCAAGAGATGG-3′, Rv: 5′-AGGAAGGAAGGCTGGAAGA-3′. All gene mRNA expression values were normalized to the expression level of the housekeeping gene (actin) and quantification of gene expression was performed using ΔCt values, defining ΔCt as the difference between the target and reference gene Ct values.

### 2.10. In-Cell Western Assays

3 × 10^3^ LN-229 cells were plated in 96-well plates, synchronized, and fixed with cold methanol (−20 °C) at 0, 3, 6, 9, 12, 15, 18, 21, or 24 HPS (3 h intervals during 24 h). Cells were then washed with phosphate-buffered saline (PBS), blocked with 5% milk and treated with either TIAM1-Alexa 488 (Santa Cruz Biotechnology, Dallas, TX, USA), BMAL1 (Novus, Centennial, CO, USA), and PER1 (Santa Cruz Biotechnology, Dallas, TX, USA) monoclonal antibodies. Measurements were performed using a Cytation 5 Imaging Reader, 10x objective and DAPI and GFP channels.

### 2.11. Image Acquisition and Processing

Raw images were acquired with the GEN 5 software in the Cytation 5 system and were processed using the Fiji image analysis software [32]. Raw images were obtained to qualitatively confirm the staining patterns, so no intensity measurements were performed. Intensity measurements were only performed in migration experiments. For each experiment, raw images were processed to aid visualization in the following way: first, both unsharp-mask and despeckle filters were applied to separate channels simultaneously. Next, all channels were merged into a single RGB color-type image. The result of this process is a final image that resembles one of the blue-green images displayed in Figure 1E,F.

For migration experiments, images were automatically acquired and stitched by the GEN 5 software (Winooski, VT, USA) and were processed with Fiji (version 1.53a, Wayne Rasband, Stapleton, NY, USA). Then, the unsharp-mask and despeckle filters were applied, a flat-field correction was used to remove grid effects generated by the stitching process.

### 2.12. Animals

The animals used throughout the study were 2-month-old, male, NIH Swiss foxN1(Δ/Δ) nude mice, which were purchased from Universidad Nacional de La Plata, Argentina. Mice were housed in group cages (5 individuals) and were kept under a 12 h light:12 h dark (LD) cycles [with zeitgeber time 12 (ZT12) defined as the time of lights OFF, i.e., local time 7 p.m., and ZT0 defined as lights ON, i.e., local time 7 a.m.], with temperature set at 22 ± 2 °C. The animals also had ad libitum access to balanced rodent chow and water. Animals were kept for one week under LD cycles. All experimental procedures with animals were approved by the Institutional Animal Care and Use Committee of the Universidad Nacional de Quilmes (project #005-16, September 2016) as established by the Guide for the Care and Use of Laboratory Animals (NIH, Bethesda, MA, USA).

### 2.13. Intracranial Surgery and Xenograft Implants

Approximately, 2 × 10^5^ cells/μL of LN229 viable cells were implanted in each nude mouse with a 33 ga syringe (Hamilton, Franklin, MA, USA). A stereotaxic device (Stoelting and Co., Wood Dale, IL, USA) was used to aim the syringe towards the right striatum. Using the Bregma as zero point, the corresponding coordinates are: ML: −2, AP: 0, DV: −3.4. All surgeries were performed between ZT3 and ZT11, and then animals were randomly assigned to each group. Once the mouse underwent the surgery, it was allowed to recover for at least seven days, receiving 6 mg/kg sub-cutaneous ampicillin and 0.05 mg/mL ibuprofen via drinking water, before the onset of the chronopharmacological protocol. 

### 2.14. Chronopharmacological Drug Administration

Mice received a daily intraperitoneal injection of the drug 1A-116 (20 mg/kg, 200 μL) or vehicle, at either ZT3 or ZT12. Currently, there are no available data allowing the extrapolation of LN299 circadian timing in vitro to in vivo conditions. Thus, two timepoints were selected for drug administration, one at ZT12 and the other one at ZT3. At ZT3 and ZT12, mice are expected to display differences in many important physiological variables including core body and distal temperature, liver and renal activity, hormonal blood levels, and metabolic profile [23]. Thus, these two timepoints serve as a good starting point to observe differences in the response to a drug. The animal’s survival time was measured twice a week. Finally, to confirm the presence of the tumor, a necropsy was performed. In this procedure, brains were dissected and fixed in a paraformaldehyde solution, thereafter, histological sections were obtained to perform cresyl violet staining.

### 2.15. Statistical Analysis

Normal distributions were tested with the Shapiro-Wilkinson Normality Test and then parametric analyses were performed (two-way and three-way ANOVA as indicated in the text) and post hoc tests (Multiple comparisons Tukey’s and Sidak tests) were applied when indicated. Results are presented as Mean ± SEM of experimental datasets.

Circular statistics analyze the time series for period detection better than linear statistics. Thus, cosine models were used to test whether variables were rhythmic or not. For data derived from in-cell western experiments and daily variation of proliferation inhibition effects of 1A-116, circadian rhythms were assessed using COSINOR fitting data with a cosine function with set parameters: a 24 h period, testing amplitude = 0 as the null hypothesis, Lomb–Scargle (LS) periodogram and JTK_CYCLE (JTK). Data were considered rhythmic when at least two independent methods found significant rhythms and their period was measured, fitting to a fixed period of 24 h. COSINOR analysis was performed using the El Temps package [33] and the Meta2d function (to run LS and JTK) was run with the Metacycle R package [34].

Bioluminescence recordings were processed and analyzed with the Chronostar 1.0 software. First, they were filtered with a 3-h moving average to remove high-frequency variations. Next, a 4th order polynomial fit was performed on the data and then subtracted from the data (detrending filter) to remove low-frequency variation. Then, a cosine model was fitted to the data to determine the period and goodness of fit (CC). Data series were considered circadian if they had a measured period within the 20 to 28 h range and a CC of over 0.85.

Survival times of tumor-bearing mice under chronopharmacological protocol were analyzed using Kaplan-Meier survival curves, which were then compared using a log-rank test. Statistical analyses were performed with GraphPad Prism 7 (GraphPad Software, La Jolla, CA, USA) and OriginPro9 software programs. Results with p-values lower than 0.05 were considered statistically significant.

## 3. Results

### 3.1. LN229 Cells Exhibit a Functional Circadian Clock That Modulates TIAM1 Expression

Most mammalian cells display circadian rhythms even when cultured in vitro. When properly synchronized, those rhythms are evident at the population level. However, rhythmicity can be severely affected or completely lost in tumoral cells due to their high mutation rates. Thus, the first step in this analysis was to assess the presence of circadian rhythms in LN229 cultures, focusing specifically on the canonical clock genes. LN229 cells showed a circadian rhythm in Bmal1 mRNA levels (Figure 1A), which showed a dampened, low amplitude cycle in LN229E1 cells (Figure 1B) (Bmal1 LN229: JTK: *p* < 0.05, LS: ns, Meta2d: *p* < 0.05, Amp: 13.35. Bmal1 LN229E1: JTK: *p* < 0.05, LS: ns, Meta2d: *p* < 0.05, Amp: 1.5) In addition, rhythmic expression of the BMAL1 promoter activity in LN229 cells was found for over 72 h in cells carrying a pGL3-Bmal1-dLuc reporter plasmid, while no circadian variations were observed in the LN299E1 line carrying the same reporter (Appendix A). Then, clock proteins BMAL1 and PER1 were studied using in-cell westerns (Figure 1C-F). Statistical analyses were performed to identify the presence of circadian rhythms in BMAL1 and PER1 expression, indicating that synchronized LN229 cultures displayed circadian rhythms (BMAL1: JTK: *p* < 0.0001, LS: *p* < 0.01, Meta2d: *p* < 0.001, PER1: JTK: *p* < 0.05, LS: ns, Meta2d: *p* < 0.05) (Figure 1C). Minimum expression levels of BMAL1 were observed at 9 HPS, with peak expression observed at 21 HPS. Rhythmic PER1 expression levels varied in antiphase with those of BMAL1, with maximum expression at 6 HPS and minimum levels at 18 HPS. These rhythms were lost in LN229E1 cell lines (BMAL1: JTK: ns, LS: ns, Meta2d: ns; PER1: JTK: ns, LS: ns, Meta2d: ns) (Figure 1D).

The rhythmic variation under constant culture conditions in mRNA and protein level of BMAL1, in addition with PER1 rhythmic expression, combined with the lack of such rhythmicity in PER1 and BMAL1 clock proteins and the dampened rhythms in Bmal1 mRNA in LN229E1 cultures, suggest that LN229 cells exhibit a functional circadian molecular clock.

Circadian modulation of TIAM1 expression could lead to time-dependent variations of the effects of drugs targeting pathways where this protein plays a role. The drug 1A-116 was previously shown to inhibit RAC1 activation in a TIAM1-dependent manner, with larger effects of the drug observed with higher TIAM1 expression levels [13,35]. Thus, the circadian modulation of Rac1 and Tiam1 mRNA levels was studied in synchronized LN229 cultures. Tiam1 showed a significant circadian rhythm, whereas no rhythm was observed for Rac1 levels (Figure 1A) (Tiam1: JTK: 24 h period, *p* < 0.05, LS: ns, Meta2d: *p* < 0.05. Rac1: JTK: ns, LS: ns, Meta2d: ns). No circadian rhythms were found in LN229E1 cells for both Tiam1 and Rac1 (Figure 1B).

In-cell Western assays also showed a significant circadian variation of TIAM1 protein expression (Figure 1C) (JTK: *p* < 0.0001, LS: *p* < 0.01, Meta2d: *p* < 0.0001), with maximum expression levels observed at 8 HPS, and minimum levels at 20 HPS, in antiphase with BMAL1 expression. No circadian rhythms were observed in LN229E1 cells with a disrupted circadian clock (JTK: ns, LS: ns, Meta2d: ns) (Figure 1D).

The results obtained for both mRNA and protein levels indicate that TIAM1 is modulated by a functional circadian clock in LN229 glioma cells, while Rac1 showed a non-circadian expression at the mRNA level.

### 3.2. 1A-116 Inhibits GBM Cell Proliferation in a Circadian Manner

The main goal of any chronopharmacological approach is to find the circadian time of maximum effectiveness of the drug under study and if possible, the time when the ratio between desired and unwanted effects is largest. Since circadian modulation of the expression of TIAM1 was observed and it was previously reported that 1A-116 inhibits the Rac1 pathway in Tiam1 overexpressing cells [13,35], we hypothesized that 1A-116 effects should display circadian rhythms related to those of circadian TIAM1 levels. To test this hypothesis, LN229 cells (Figure 2A) were treated with either vehicle or 20 μM 1A-116 at 5 h intervals over 25 h (each culture was treated only once).

At the time of measurement, vehicle-treated cultures were close to becoming confluent monolayers, unlike the ones treated with 1A-116. Proliferation was inhibited by 20 μM 1A-116 in a circadian-dependent manner, with maximum inhibitory effects observed at 10 HPS, minimum at 23 HPS, and significant circadian periodicity (JTK: *p* < 0.01, LS: *p* < 0.05, Meta2d: *p* < 0.01) with a maximum phase approximate to the one observed for maximal TIAM1 expression (see Figure 1C).

To further study the circadian response to 1A-116, dose-response curves were performed for LN229 and LN229E1 cells at 10 HPS and 23 HPS (Figure 2B), the times of maximum and minimum 1A-116 inhibition of proliferation, respectively (see Figure 2A). In LN229 cells, a significant time dependency was observed for both 10 and 20 μM 1A-116, with larger effects observed in cultures treated at 10 HPS. In LN229E1 cells, intermediate IC50 values compared to those observed for LN229 at both 10 and 23 HPS were found, but with no significant circadian-dependency at both 10 and 20 μM. The inhibitory effects of 1A-116 on proliferation reached saturation at doses higher than 20 μM in all groups. In LN229 cells the IC50 evidenced significantly higher drug efficacy at 10 HPS, when compared to 23 HPS, while this difference was not observed in LN229E1 cells. IC50 values ranged from 10.93 ± 0.9 μM for LN229 cultures treated at 10 HPS, to 30.85 ± 1.78 μM for LN229 cultures treated at 23 HPS (Figure 2C).

### 3.3. Pro-Apoptotic Effects of 1A-116 Are Circadian-Dependent

Slowing down the proliferation of tumoral cells is important for cancer treatments, but inducing cell death is critical. Apoptosis is a regulated process of cell death that can be distinguished by several morphological and biochemical changes, including fragmentation of nuclear chromatin and loss of membrane asymmetry [36]. One of such changes is the translocation of phosphatidylserine from the cytoplasmic to the external surface of the cell membrane. Thus, detecting phosphatidylserine on the cell surface with fluorophore-labeled Annexin V (a protein with a high affinity for phosphatidylserine) [37] is a widely used method to measure apoptosis, reporting fluorescence intensity [38].

It was previously reported that 1A-116 can induce apoptosis [13], so the dependence on the time of treatment of this effect was tested in GBM cells under 6 h treatment with 20 or 50 μM 1A-116 (Figure 3). When applying 20 μM 1A-116 at 10 HPS or 23 HPS, a significant difference in the Annexin V fluorescence signal was observed, with higher levels detected at 10 HPS indicating more cells were undergoing apoptosis in the cultures treated at 10 HPS than in those treated at 23 HPS. At 50 μM, 1A-116 apoptosis induction showed no statistical differences when comparing cultures treated at 10 HPS with those treated at 23 HPS. Similar to the results obtained for proliferation assays, the pro-apoptotic activity of 1A-116 20 μM was more effective when applied at 10 HPS, than when applied at 23 HPS (Figure 3A,B).

### 3.4. 1A-116 Inhibits LN229 Cell Migration Only at 10 HPS

GBM tumors are highly invasive, making complete surgical removal and radiotherapy extremely difficult and leading to recurrence of the disease due to the survival of GBM cells outside of the treated area. Thus, beyond killing and preventing the proliferation of tumoral cells, inhibition of migration is another desirable characteristic for GBM treatments. To assess the time dependence of the effects of 1A-116 on cell migration, the invasion of a cell-free area in cultures treated with 1A-116 or vehicle was measured at 10 HPS and 23 HPS (Figure 4).

The addition of 10 μM 1A-116 at 10 HPS significantly reduced cell migration when compared to cultures treated with the same concentration of 1A-116 at 23 HPS or with vehicle-treated cultures at either 10 HPS or 23 HPS. Addition of 1A-116 at 23 HPS did not affect cell migration when compared to vehicle-treated cultures, unmasking a temporal dependence of the effects of 1A-116 on cell migration. The selected 1A-116 concentration (10 μM) showed no effects on cell viability (Appendix A). Additionally, no significant differences were found in the number of cells outside of the cell-free area, but since this variable can affect the total number of migrating cells, they were quantified to normalize the migration data (Appendix A) to take into account possible differences in the number of cells available to migrate into the cell-free area.

### 3.5. Daytime Dependent Effects of 1A-116 Treatment in Survival of GBM Nude Mice Model

The results obtained in vitro demonstrate that the effectivity of 1A-116 is modulated by a circadian oscillator present in LN229 cells and support testing in vivo the daytime-dependent effect of 1A-116 drug delivery in a murine GBM model. To this end, LN229 cells xenografts were implanted in the brain of nude mice to generate gliomas.

1A-116 was administered at two different circadian times corresponding to ZT3 and ZT12 (i.e., 3 h after lights on and at the time of lights off, respectively). When comparing the effects of injecting 1A-116 or vehicle at ZT3 vs. ZT12, a significant increase in survival time was observed when 1A-116 was administered at ZT12 compared both to the same treatment at ZT3 and to vehicle groups. The median survival times obtained were 73 days for 1A-116 at ZT12, 68 days for ZT3, 64.5 days for the vehicle at ZT12 and 63.5 days for the vehicle at ZT3 (Mantel–Cox log-rank test, *p* < 0.05) (Figure 5), demonstrating that the effectivity of the 1A-116 can be improved only by varying the time of administration. These results suggest that a chronopharmacological delivery with 1A-116 is a feasible strategy to improve survival for GBM.

## 4. Discussion

In this work, we report a circadian modulation for the effects of 1A-116 on GBM cells in vitro and on tumor-bearing nude mice in vivo. We found that LN229 cells express a functional circadian clock, displaying circadian oscillations of BMAL1, both at mRNA and protein levels, and of PER1 protein. In addition, while Rac1 mRNA showed no circadian oscillations, the GEF activator TIAM1 presented circadian rhythms both in mRNA and protein expression. Oscillations of BMAL1, PER1 and TIAM1 protein expression, as well as Tiam1 mRNA, were abolished by knocking down the *bmal1* clock-gene, while mRNA oscillations of Bmal1 were seriously impaired, indicating that the observed circadian oscillations were generated by this canonical component of the molecular circadian clock in LN229 cells. Our results show that the effects of 1A-116, a drug designed to specifically inhibit the Rac1 signaling pathway by preventing its interaction with TIAM1, were also modulated by the circadian clock in LN229 cells in vitro and showed a time-of-day dependence in a glioma tumor model in vivo. Interestingly, all the antitumoral effects studied of the drug, including proliferation inhibition, apoptosis induction and migration inhibition, were in phase with TIAM1 high levels of expression.

1A-116 was obtained by rational design from an analog selected through a docking-based virtual screening approach. It specifically blocks the interaction site of RAC1 with its GEFs, particularly RAC1-TIAM1 and RAC1-DOCK180, and it was reported to induce apoptosis and inhibit proliferation and migration in LN229 and U-87 glioma cell lines [13]. Considering that this drug was proposed as a good candidate to treat GBMs and other cancers, we decided to study the circadian modulation of the effects of 1A-116 on critical cellular processes for cancer progression. Cell proliferation inhibition, apoptosis induction and cell migration were studied after the addition of 1A-116 at different times after circadian synchronization. All of these responses were dependent on the time of administration, with larger effects at the time when TIAM1 and PER1 levels are high, and BMAL1 levels are low. When exposing cells to 10 and 20 μM of 1A-116, proliferation was inhibited when the drug was applied at 10 HPS, but not at 23 HPS, while no circadian dependency was observed at higher, saturating concentrations. Accordingly, an IC50 reduction on cell proliferation by 1A-116 was observed when cultures were treated at 10 HPS when compared to 23 HPS. This difference was absent in LN229E1 cells lacking circadian rhythms of BMAL1, TIAM1 and PER1 expression, suggesting that the efficacy of the drug is regulated by the circadian molecular clock. When applied close to the time of TIAM1 peak of expression, low concentrations of 1A-116 (20 μM) elicited similar effects in LN229 cells, to those obtained at other time points with saturating concentrations. Similar to the results observed for proliferation inhibition, apoptosis was induced at 20 μM by 1A-116 when it was applied at 10 HPS, but not when added at 23 HPS. In addition, no effect on cell viability was evident at lower concentrations, 5 and 10 μM, of 1A-116 at both 10 and 23 HPS. This low-ranged 10 μM dose was efficient to inhibit cell migration when applied at 10 HPS, but not when applied at 23 HPS, confirming the circadian dependency of the effects of 1A-116 also in this process.

When 1A-116 was applied on tumor-bearing mice, it was more effective at extending survival time when delivered at ZT12 than at ZT3. Many anti-cancer drugs have already been shown to decrease tumor growth and extend survival when applied at different circadian times both in mice and humans [22,23,26]. Chronotherapy presents an interesting treatment option, allowing us to improve the effectiveness of the therapies just by picking the optimal time of application, as shown by treatment with 1A-116 in our glioma model. The current therapy of choice for GBM is TMZ, which prolongs patient survival by just a few months on average [39] and its effects were also recently shown to be dependent on the time of day at which it is administered [26]. It is necessary to continue studying new drugs and treatment schedules that may extend life expectancy and overall survival for patients, ideally with fewer and milder effects. In this regard, our results indicate that 1A-116 shows potential as a new drug for chronopharmacological GBM treatment and supports further studies in combination with TMZ.

Much evidence links the circadian system to cancer risk and progression [40]. Disrupted circadian rhythms were reported in different cancer models [41] and several lines of evidence link glioma progression to the circadian clock [42,43]. Two genes that play important roles in circadian clock regulation, casein kinase-1 epsilon (CK-1ε) and the nuclear receptor NR1D2, were found to regulate GBM cell survival and were proposed as targets for glioma treatment [44]. Additionally, the P38 MAPK pathway, linked to circadian entrainment [45] was proposed as a good target to treat different cancers including GBM, and found to be under circadian control in normal astrocytes, but deregulated in GBM cells [46]. In the context of this work, what makes the RAC1 pathway an interesting target from a chronopharmacological perspective, is that it is related to most of the pathways linking GBM to the circadian clock: (a) an NR1D2 knock-down was found to impair *p*-Rac1, (b) the p38 MAPK pathway is known to be activated by RAC1 and (c) CK-1e mutants were shown in breast cancer to activate the non-canonical Wnt/Rac-1/JNK pathway contributing to cancer development [47].

Taking a chronopharmacological approach in a given treatment focuses on finding the optimal time for drug application. Defining an optimal time is non-trivial since it depends on a large number of processes that need to be taken into consideration, including desired and side effects, costs, the feasibility of proposed schedules, tolerance, pharmacodynamics and pharmacokinetics [23]. However, in most cases, the optimal time for treatment will be the one generating the maximum therapeutic index, the largest ratio between desired and side effects [48]. The chronotherapeutics of anti-tumor drugs were tested on a wide variety of cancers, finding that the time of administration can generate significant differences in survival and tumor growth [49]. Levi et al. showed that the administration of oxaliplatin with a chronopharmacological scheme in patients with colorectal cancer could reduce its toxicity on normal cells [22]. For GBM, temozolomide and VX-745 (a p38 MAPK inhibitor) effects were reported to depend on the time of administration in vitro [46,50]. In our model, higher efficacy of 1A-116 was observed at circadian times of low BMAL1 expression in GBM cells and a differential overall survival was found when applying 1A-116 at ZT12 to glioma-bearing nude mice. These results suggest that a chronopharmacological application of 1A-116 is a feasible strategy to improve survival. Further studies need to be performed to identify the pharmacological mechanisms underlying the in vivo time dependency of 1A-116 treatment as well as to find the optimal treatment time, assessing administration with 1A-116 in more Zeitgeber times. In addition, future research should focus on characterizing the circadian clock of GBMs in vivo, in order to determine the regulation of the circadian clock on 1A-116 effectivity. Taking into account that the circadian clock of tumors in vivo may be disrupted, the use of a chronobiotic, (such as melatonin) [51], could be an interesting strategy to reset the circadian staging of targets of the drug. Characterizing the rhythmicity of GBM tumors in vivo will require complex experimental approaches, probably using advanced bioluminescence and fluorescence imaging techniques, and translating those results to humans is likely to pose even more difficult challenges, but considering the urgent need to improve GBM therapies, these are efforts worth undertaking.

The chronotherapy of cancer medications thus far has been based on animal model studies, and potentially represents the combined effects both of chronopharmacokinetic and chronopharmacodynamic phenomena. The findings of this work are representative of the chronopharmacokinetics and chronopharmacodynamics of 1A-116 in combination and can be the basis for human trials to improve outcomes of this very aggressive and difficult-to-survive cancer. Thus, our results provide a good starting point once the circadian phase relationship between tumors and host clocks is defined.

Since circadian expression of TIAM1 had not been reported before and Rac1 mRNA levels were found not to be under circadian control in this work and in [52], circadian modulation of the RAC1 pathway could have easily been overlooked. Further research should look into the mechanisms and implications linking the circadian molecular clock and the entire Rac1 pathway. In addition to our results, previous data on rhythmic expression of TIAM1 can be found only in the CircaDB database for lung samples [53], indicating that further work should determine whether this modulation occurs in other tumoral and normal cells and tissues, the mechanisms underlying its circadian control, and if it is relevant for treatment.

Circadian oscillations of 1A-116 effectivity may be the consequence of a variety of factors, including circadian modulation in the expression of its molecular target, modulation of upstream and downstream components of the affected pathways, circadian control of the transport of 1A-116 into the cells, metabolization, and even cellular processes making the cells more prone to enter apoptosis or slow down cell proliferation. Importantly, a reciprocal interaction between components of the circadian clock and the cell cycle has also been demonstrated. The circadian gene *bmal1* induces the expression of the cell cycle genes *wee* [54], cyclins B and D [55] and *p21* [56] and some clock genes are expressed at specific phases of the cell cycle [57], demonstrating this molecular cross-talk [58]. Recent studies reported that either decreasing or enhancing circadian clock function can inhibit tumor cell proliferation [59,60]. Further studies are necessary to determine if the circadian modulation of the effects of 1A-116 is in any way related to the cell cycle phase. It was previously shown that the circadian rhythms of pharmacokinetic processes play an important role in the temporal variations in the effectivity of several drugs [61,62]. Therefore, future studies involving 1A-116 should consider a chronopharmacological approach, addressing both chronopharmacodynamics (i.e., circadian interaction with RAC1) and chronopharmacokinetics, to fully unveil the mechanisms underlying the modulation of its activity. Interestingly, in the last decades, targeted therapies have provided a novel approach to treat cancer. Our results show that further refinement can be achieved considering circadian oscillations in the expression of drug targets and associated pathway components. The correct timing of drug dose has been a largely underappreciated factor in established therapeutic schemes for most diseases [63], improving therapeutic outcomes as in hypertension [64]. This becomes particularly important in cancer treatment, where the therapeutic window of most drugs is generally narrow and there is a need to optimize therapeutic outcomes minimizing the associated side effects [65]. This work highlights the opportunity to exploit a chronotherapeutic approach with an agent that blocks Rac1 interaction with its GEFs, providing promising results to tackle glioblastoma, an aggressive cancer with extremely limited therapeutic options.

## 5. Conclusions

In LN229 cells with a functional circadian clock, we observed the circadian expression of the protein TIAM1, together with the circadian-dependent response to the novel chemotherapeutic agent 1A-116. The circadian dependency was lost in LN229E1 cells in which the *bmal1* clock gene was knocked down, supporting the hypothesis of the circadian control of the response to 1A-116. In our glioma-bearing mice, 1A-116 administered at ZT12 increased the survival time when compared both to animals treated with 1A-116 at ZT3 and those treated with vehicle.

These data unveil the circadian modulation of TIAM1, one of the main GEF activators of RAC1 and the chronomodulation of 1A-116. Our results should be taken into consideration in future preclinical studies for glioblastoma treatment, as well as in all studies related to GEFs and the RAC1 signaling pathway.

## Figures and Tables

**Figure 1 pharmaceutics-13-01091-f001:**
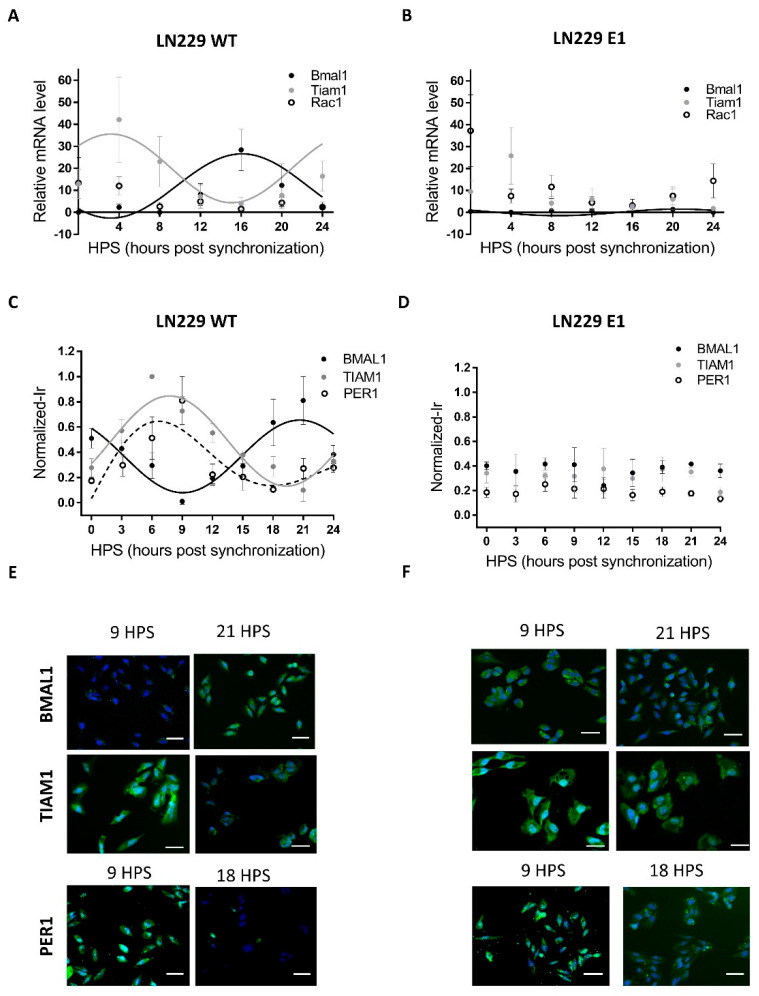
Circadian mRNA levels and protein expression of Bmal1, Per1, Rac1 and Tiam1 in LN229 and LN229E1 cells. (**A**,**B**) After synchronization with FBS, cells were fixed at 4-h intervals, and their mRNA was extracted. qPCR was done with Tiam1, Rac1 or Bmal1 primers, to determine the amount of Tiam1, Rac1 or Bmal1 levels relative to housekeeping (actin) mRNA levels. (**A**) In LN229 cells, Bmal1 and Tiam1 mRNA levels show circadian rhythms in antiphase. When fitting to a fix 24 h period, their p-values were: Tiam1: *p* < 0.05, Bmal1: *p* < 0.05, Rac1: ns. (**B**) Contrary to the previous sample, no circadian rhythms were found in the LN229E1 sample for Rac1 and Tiam1, which had measured *p*-values of *p* > 0.05. Circadian rhythms were still found for Bmal1, but with a decreased amplitude. (**C**,**D**) After synchronization with FBS, cells were fixed at 3-h intervals, processed for Tiam1, Per1, or Bmal1 immunocytochemistry (green channel), and nuclei counterstained with DAPI (blue channel) (**C**) BMAL1, PER1, and TIAM1 normalized immunoreactivity was circadian and in antiphase in LN229. When fitting to a fixed 24-h period, their *p*-values were: TIAM1: *p* < 0.0001, BMAL1: *p* < 0.0001, PER1: *p* < 0.05. (**D**) No circadian rhythms were observed in LN229E1, which had a *p*-value of *p* > 0.05. (**E**,**F**) Representative images obtained with the Cytation 5 fluorometer system, which was used to qualitatively confirm the staining results in LN229 and LN229 E1 cells. The scalebars represent 20 μm.

**Figure 2 pharmaceutics-13-01091-f002:**
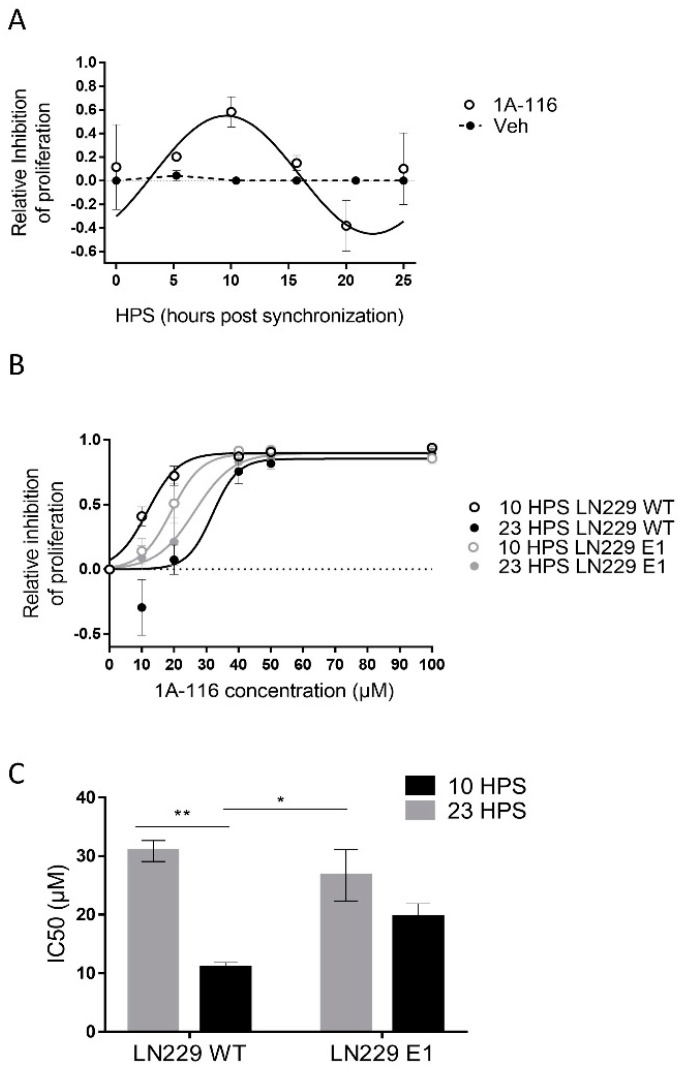
Circadian variation in the effect of 1A-116 on proliferation in the LN229 glioblastoma cell line. (**A**) Proliferation inhibition relative to vehicle-treated cells. The assessment was conducted with crystal violet 0.1% staining, 72 h after applying the treatment of 20 μm of 1A-116 or vehicle at different HPS. Circadian modulation of 1A-116 effects on proliferation was observed with significant values when fitting to a fix 24 h period: Meta2d: *p* < 0.01, two-way ANOVA, ns. (**B**) Dose–response curves for proliferation inhibition relative to vehicle-treated cells in LN229 and LN229E1 cultures, which were treated at 10 HPS or 23 HPS (mean ± SEM). A larger response was observed when LN229 was treated at 10 HPS; Three-way ANOVA: HPS, *p* < 0.0001; 1A-116 Concentration, *p* < 0.0001; 1A-116 Concentration × HPS, *p* < 0.0002; HPS × Cell type, *p* < 0.0041; 1A-116 Concentration × HPS × cell type, *p* < 0.05, N = 3. Only main comparisons are stated for results obtained with Tukey´s multiple comparisons test: LN229 WT 10 HPS 10 μM vs. LN229 WT 23 HPS 10 μM, *p* < 0.0001, LN229 WT 10 HPS 20 μM vs. LN229 WT 23 HPS 20 μM, *p* < 0.0003; LN229 WT 10 HPS 20 μM vs. LN229E1 23 HPS 20 μM, *p* < 0.013, LN229 WT 10 HPS 0 μM vs. LN229 WT 10 HPS 20 μM, *p* < 0.0001, LN229E1 10 HPS 0 μM vs. LN229E1 10 HPS 20 μM, *p* < 0.026). The dose-response curves were obtained by fitting the median data values, which were calculated from three separate experiments. (**C**) IC50 values were obtained for independent experiments. It shows significantly lower IC50 values for LN229 cultures treated at 10 HPS. No time dependency was found for LN229E1. The following values were obtained: Two-way ANOVA, for factor HPS, *p* < 0.001, HPS x cell line, *p* < 0.05. Tukey’s multiple comparisons test reported: 10 HPS LN229 vs. 23 HPS LN229 ** *p* < 0.01,10 HPS LN229 vs. 23 HPS LN229E1, * *p* < 0.05. All plotted data represent mean ± SEM.

**Figure 3 pharmaceutics-13-01091-f003:**
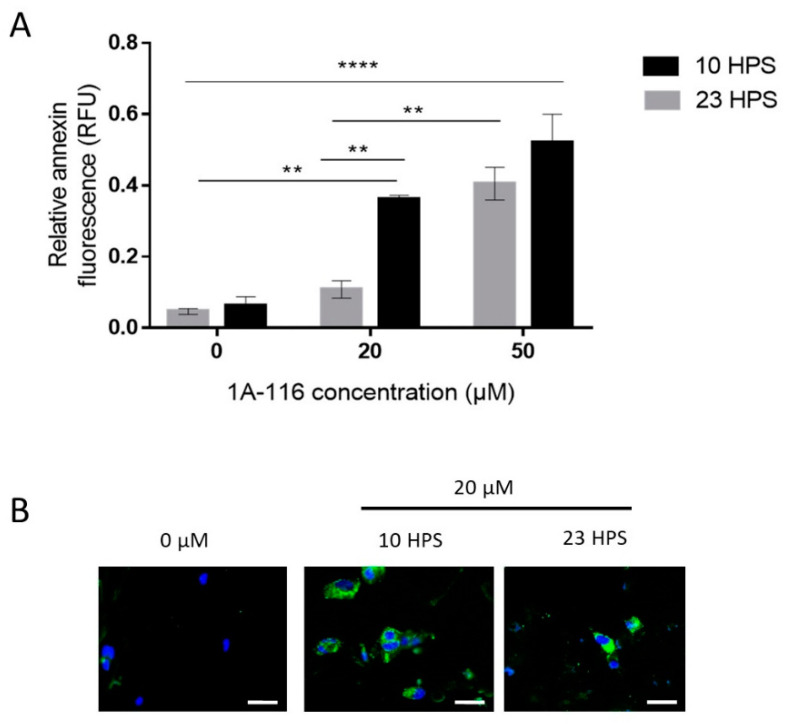
Apoptosis induction by 1A-116 is higher at 10 HPS in LN229 cells. Annexin V staining was used to measure early apoptosis in LN229 cultures 6 h after the addition of vehicle, 20 μM, or 50 μM of 1A-116. Fluorescence intensity was measured in a Cytation 5 system and DAPI staining was used to relativize the number of cells in each well. (**A**) Relative early apoptosis values were obtained for three independent experiments (N = 3) and were compared between treatments at 10 HPS and 23 HPS. Significantly larger values were observed when the treatment was applied at 10 HPS (two-way ANOVA, for factor HPS *p* < 0.01, factor Concentration *p* < 0.001, interaction *p* < 0.05. Multiple comparisons Tukey´s tests: 0 μM 23 HPS vs. 20 μM 10 HPS: ** *p* < 0.01, 0 μM 10 HPS vs. 20 μM 10 HPS: ** *p* < 0.01, 0 μM 23 HPS vs. 50 μM 23 HPS: ** *p*< 0.01, 0 μM 10 HPS vs. 50 μM 10 HPS: **** *p* < 0.0001, 0 μM 23 HPS vs. 50 μM 10 HPS: **** *p* < 0.0001, 20 μM 23 HPS vs. 20 μM 10 HPS: ** *p* < 0.01, 20 μM 23 HPS vs. 50 μM 23 HPS: ** *p* < 0.01).Column values represent mean ± SEM. (**B**) Representative images that were obtained to qualitatively confirm the staining results in LN229 cells at 10 HPS and 23 HPS, 6 h after treatment with 1A-116. White scale bars represent 20 μm.

**Figure 4 pharmaceutics-13-01091-f004:**
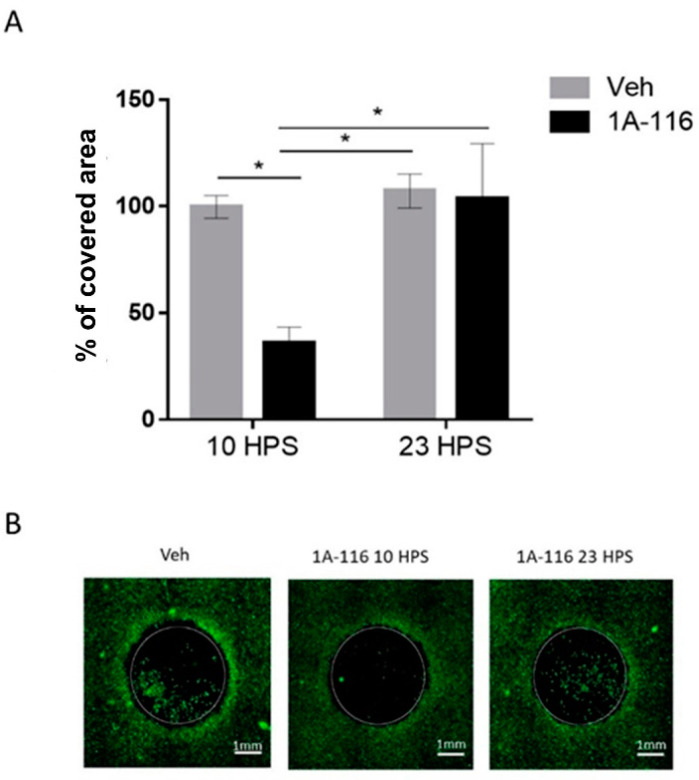
Cell migration inhibition due to 1A-116 at 10 HPS. Cell migration into a cell-free area was measured in cultures treated with 1A-116 (10 μM) or vehicle at either 10 HPS or 23 HPS. At 16h after treatment, live cells were stained with Calcein-AM. The area covered by cells in the previously cell-free area was measured with ImageJ software. The total covered area by the cells was normalized to the covered area of the vehicle (control), after the area outside the cell-free area and technical controls (t0) were subtracted out from the data. (**A**) The covered area for 10 μM treated cells was significantly reduced at 10 HPS than at 23 HPS (Two-way ANOVA, for factor HPS, * *p* < 0.05, for factor concentration, * *p* < 0.05. Sidak’s multiple comparisons test: 1A-116 10 HPS vs. 23 HPS: * *p* < 0.05; 10 HPS Veh vs. 1A-116: * *p* < 0.05, 10 HPS 1A-116 vs. 23 HPS Veh: *p* < 0.05, N = 4). Column plots represent the mean ± SEM. (**B**) Representative images of the migration area at 16 h after treatment. The white scale bar represents 1 mm, and the white circles represent the originally cell-free area where cell migration was quantified.

**Figure 5 pharmaceutics-13-01091-f005:**
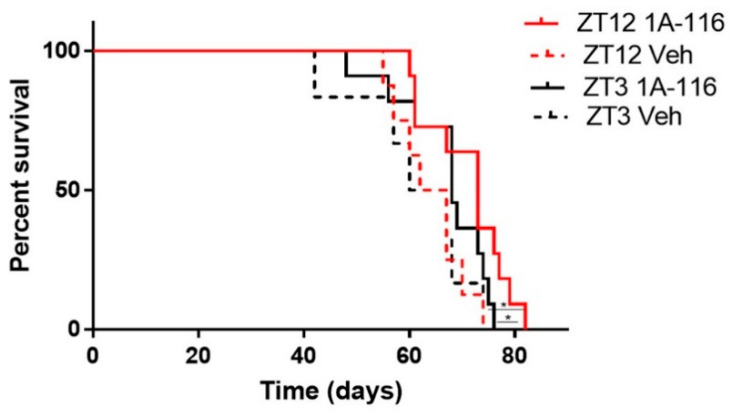
GBM bearing nude mice treated with 1A-116 at ZT12 exhibited increased survival rates than mice treated with 1A-116 at ZT3. Survival curves of nude mice treated with vehicle or 1A-116 in ZT3 and ZT12 were significantly different (*n* = 11, Mantel–Cox log rank Test, * *p* < 0.05). Median survival increased in ZT12. The median survival times obtained from fitting the data are listed: ZT12 Veh: 64.5 days, ZT3 Veh: 63.5, ZT12 1A-116: 73 days, ZT3 1A-116: 68 days.

## Data Availability

The datasets used and/or analyzed during the current study are available from the corresponding author on reasonable request.

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
