# Peer review of "Timing of Novel Drug 1A-116 to Circadian Rhythms Improves Therapeutic Effects against Glioblastoma"

_pharmaceutics, 2021, doi:10.3390/pharmaceutics13071091_

Round 1

Reviewer 1 Report

General Comments

  1. This is well conceived and conducted circadian research.
  2. However, the current wording of the title “Chronomodulated Delivery Improves the Effects of the Novel Drug 1A-116 in a Gliobastoma Model” is misleading. Chronomodulation with respect to the published literature (see: [1] Scheving LE, Burns ER, Pauly JE, Halberg F, Haus E. Survival and cure of leukemic mice after circadian optimization of treatment with cyclophosphamide and 1-beta-D-rabinofuranosylcytosine. Cancer Res. 1977 Oct;37(10):3648-55. PMID: 908013; [2] Lévi FA, Zidani R, Vannetzel JM, Perpoint B, Focan C, Faggiuolo R, Chollet P, Garufi C, Itzhaki M, Dogliotti L, et al. Chronomodulated versus fixed-infusion-rate delivery of ambulatory chemotherapy with oxaliplatin, fluorouracil, and folinic acid (leucovorin) in patients with colorectal cancer metastases: a randomized multi-institutional trial. J Natl Cancer Inst. 1994 Nov 2;86(21):1608-17. doi: 10.1093/jnci/86.21.1608. PMID: 7932825) refers to the tailoring to circadian rhythm determinants of medications when infused, e.g., by programmable infusion pumps or released by special oral or other drug-delivery dosage forms so the concentration of one or more medications differs predictably-in-time during the 24 hours.

The same conceptual mistake is found in Line 30 of the abstract and elsewhere as listed below.  

Perhaps, a more appropriate title might be: “Timing of Novel Drug 1A-116 to Circadian Rhythms Improves Therapeutic Effects against Glioblastoma”

  1. Minor comments:
  2. There are inconsistencies in the text in the use a comma before the word ‘and’ when a series of nouns are expressed (see, e.g., line 41 vs. lines 54/55 and lines 63/64)
  3. The number ‘2.5x103‘ is sometimes expressed in European format with comma instead of decimal ‘2,5x103’ (e.g., line 123) as is the ‘2,5x objective (line 172).

Specific Comments

Line 24 (Abstract): Authors use the phrase “two times of day”. Since this research specially pertains to internal circadian time, this reviewer suggests the phrasing should be more specific to internal time, something like “two circadian times”.   “Time of day” phrasing is also used elsewhere in the text (e.g., line 49, line 340) when “circadian time” seeming is much better phrasing.     

Line 149: A few words at this stage of the manuscript seem necessary to justify to the reader the basis for the selection of 10 HPS and 23 HPS to conduct the dose-response and other experiments described thereafter.

 Line 157: What is the rationale for measuring cell viability, i.e., number of fluorescent cells’, as well as cell migration and apoptosis 16 h – rather than say 14 h,  24 h or whenever -- after drug treatment?

Line 228: For how long were the mice kept under the 12L:12D cycle before use? The reader wants to know that it was sufficiently long enough to ensure synchronization of the animals circadian time structure to the imposed LD schedule.

Line 242: Please provide justification to the reader for the selection of ZT3 and ZT12 for the timing of the daily intraperitoneal injection of 1A-116 (20 mg/kg, 200 μl) and vehicle.

Line 243: The authors state that body weight was measured twice a week. Why are the findings of the body weight analysis not presented in the paper? One would expect poorer body weight maintenance (weight loss) from baseline in those treated at ZT3.

Line 455: What circadian time was the xenografts of LN229 cells implanted in the brain of the nude mice to generate gliomas? Were all of the xenografts implanted at the same circadian time or different circadian times? Does the circadian time of implantation pf the LN229 cells possibly result in different intensity of glioma generation that might confound the investigated effect of the circadian timed 1A-116 or vehicle?

Line 457: ‘Two different times of day’; why not ‘two different circadian times’?

 Lines 464-466: Poor use of terminology, i.e., ‘right time of day’ and ‘chronomodulated delivery with 1A-116’.

Line 461: Is the outcome variable used for survival time correctly termed? Is it median survival time per se or should it be the mean duration of survival from baseline when for the 4 different treatment groups combined attain 50% survival?

Line 485: ‘time-of-day’ phrasing.

Lines 517 and 522: ‘times of day’ phrasing.

Line 514: The paragraph commencing on line 514 addresses chronotherapy as an interesting treatment option. How is the term chronotherapy used here – selection of the single best circadian time to administer the medication to optimize treatment benefit or minimize adverse effects, or does it mean the best infusion profile – sinusoidal or otherwise during the 24h, with the maximum and minimum drug concentration timed to circadian determinants of medication tolerance and efficacy, or does it mean both options?      

Lines 561-569: The chronotherapy of cancer medications thus far has been based on animal model studies, like those done by the authors with 1A-116, and which potentially represent the combined/integrated effects both of chronopharmacokinetic and chronopharmacodynamic phenomena, and unless this reviewer is over-generalizing the presented work the findings of their animal study are representative of the chronopharmacokinetics and chronopharmacodynamics of A1-116 in combination and can be the basis for human trials to improve outcomes of this very aggressive and difficult-to-survive cancer.

Finally, the authors express the possibility that circadian disruption could be an aspect of cancer and perhaps make more difficult the optimal circadian timing of medications. The option as discussed at earlier chronobiology meetings is to use a chronobiotic, perhaps melatonin, to set or reset the circadian staging of target(s) of the drug chronotherapy. 

References

#20 ought to be replaced by the updated and even more impressive Hygia chronotherapy Project outcomes

Hermida RC, Crespo JJ, Domínguez-Sardiña M, Otero A, Moyá A, Ríos MT, Sineiro E, Castiñeira MC, Callejas PA, Pousa L, Salgado JL, Durán C, Sánchez JJ, Fernández JR, Mojón A, Ayala DE; Hygia Project Investigators. Bedtime hypertension treatment improves cardiovascular risk reduction: the Hygia Chronotherapy Trial. Eur Heart J. 2020 Dec 21;41(48):4565-4576. doi: 10.1093/eurheartj/ehz754. PMID: 31641769.

Other recent publications for consideration are:

 Hermida RC, Hermida-Ayala RG, Smolensky MH, Mojón A, Fernández JR. Ingestion-time differences in the pharmacodynamics of hypertension medications: Systematic review of human chronopharmacology trials. Adv Drug Deliv Rev. 2021 Mar;170:200-213. doi: 10.1016/j.addr.2021.01.013. Epub 2021 Jan 22. PMID: 33486007.

Hermida RC, Mojón A, Hermida-Ayala RG, Smolensky MH, Fernández JR. Extent of asleep blood pressure reduction by hypertension medications is ingestion-time dependent: Systematic review and meta-analysis of published human trials. Sleep Med Rev. 2021 Jan 23;59:101454. doi: 10.1016/j.smrv.2021.101454. Epub ahead of print. PMID: 33571840.

Reviewer 2 Report

Currently, a large gap exists between solid theoretical background for the benefits of the chronotherapeutic approaches and the paucity of the original papers that clearly demonstrate it. Chronomodulation of therapy that targets specific properties of the malignant cells is one of such rare cases. This valuable paper adds novel findings to the use of proper timing to gain maximum therapeutic benefits with a newly developed drug, 1A-116 for the treatment of glioblastoma.

Validation of complex antitumoral effects of the drug (proliferation/migration inhibition, apoptosis induction) that in synchrony with TIAM 1 maximum expression, is most interesting.

Paper is recommended for publication, though, there are several, mostly methodical questions:

  1. Results, and Figure 1. While presenting and discussing results on the circadian rhythms in mRNA levels and protein expression of Bmal1, Per1, Rac1, and Tiam1 in LN229 and LN229E1 cells, different in exact period (best-fit harmonics?) of the circadian range, are considered. These circadian periods vary from shorter to longer than 24 hours, with no apparent reasoning for such discrepancy.
  2. Thus, another question arises, what was the sampling of the time-series used for cosinor (how many consecutive cycles it included, and what were intervals between samples, 4 hours in each case)? It is not clearly described in the Methods.
  3. It is most likely that different periods were just a statistical artifact from the limited density (6 points per day) and duration (how long it was?) of the time series, with no obvious physiological meaningfulness. Why did the authors decide to calculate and compare best-fit periods around 24 hours, instead of fixed tau=24h? Regarding sampling, it is quite unlikely, that any of these periods were significantly different from tau=24 hours if one will consider confidence intervals for the estimated tau.
  4. An alternative approach to validate the presence (or absence) of the 24-h rhythms in substances/cell lines could be to use normalized instead of the raw values of protein and RNA expression. This could help much in cases when mean expression values vary considerably among individual time series.
  5. While discussing in Vivo results on effects of time of the day on 1A-116 treatment in the survival of GBM nude mice, authors selected ZT3 and ZT12. Why exactly this time, what was physiological reasoning for choosing it, also since these time-points are not equidistant in terms of 24 cycle, or light regimen? Could it be that effect is even better at some other time point?

Author Response

Please see the attachment,
